# Random Multi-Channel Image Synthesis for Multiplexed Immunofluorescence Imaging

## Abstract

Multiplex immunofluorescence (MxIF) is an emerging imaging technique that produces the high sensitivity and specificity of single-cell mapping. With a tenet of "seeing is believing", MxIF enables iterative staining and imaging extensive antibodies, which provides comprehensive biomarkers to segment and group different cells on a single tissue section. However, considerable depletion of the scarce tissue is inevitable from extensive rounds of staining and bleaching ("missing tissue"). Moreover, the immunofluorescence (IF) imaging can globally fail for particular rounds ("missing stain"). In this work, we focus on the "missing stain" issue. It would be appealing to develop digital image synthesis approaches to restore missing stain images without losing more tissue physically. Herein, we aim to develop image synthesis approaches for eleven MxIF structural molecular markers (i.e., epithelial and stromal) on real samples. We propose a novel multi-channel high-resolution image synthesis approach, called pixN2N-HD, to tackle possible missing stain scenarios via a high-resolution generative adversarial network (GAN). Our contribution is three-fold: (1) a single deep network framework is proposed to tackle missing stain in MxIF; (2) the proposed "N-to-N" strategy reduces theoretical four years of computational time to 20 hours when covering all possible missing stains scenarios, with up to five missing stains (e.g., "(N-1)-to-1", "(N-2)-to-2"); and (3) this work is the first comprehensive experimental study of investigating cross-stain synthesis in MxIF. Our results elucidate a promising direction of advancing MxIF imaging with deep image synthesis.

**Keywords:** MxIF  GAN  Multi-channel  Multi-modality

## 1. Introduction

Inflammatory bowel disease (IBD), such as Crohn's disease (CD), leads to chronic, relapsing and remitting bowel inflammation Baumgart and Sandborn (2012), with high prevalence (3.1 million Americans) Dahlhamer et al. (2016). The ∗ ∗ ∗ ∗ ∗∗ project, with Multiplexed Immunofluorescence (MxIF) collected [∗ ∗ ∗ ∗ ∗∗], provides the unique opportunity of mapping cell number, distribution, and protein expression profiles as a function of the anatomical location of IBD. MxIF is an emerging imaging technique that stains and scans extensive numbers of antibodies iteratively, providing comprehensive biomarkers to segment and group different cells from a single tissue section Lin et al. (2015); Stack et al. (2014).

However, the unprecedented rich information at the cellular level via MxIF is accompanied by new challenges for imaging. In this work, we employ a well-established staining protocol employed for years with static stain-wash cycles to collect the MxIF data [∗∗∗∗∗∗]. One well-known challenge is the considerable depletion of the scarce tissue via extensive rounds of staining and bleaching (called the "missing tissue" problem). Moreover, the Immunofluorescence (IF) imaging can globally fail for particular rounds (called the "missing stain" problem) (Fig. 1), which is an extreme case of missing tissue (occurs 3% of the chance as the empirical observation). In this study, we focus on the "missing stain" problem . It is often impractical to recover the missing pieces of information physically. However, it

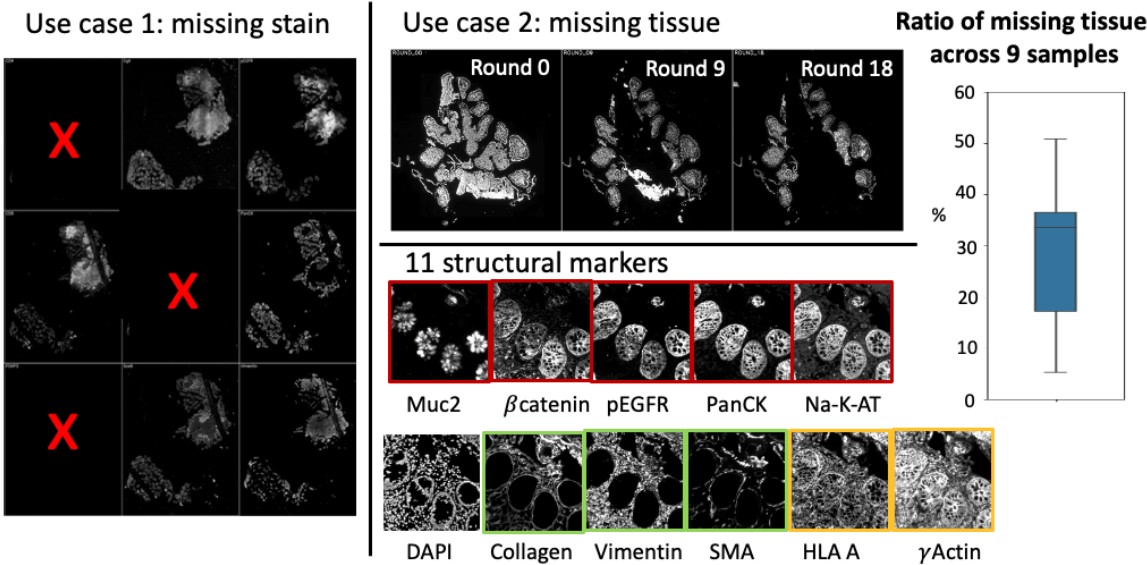

Figure 1: There are two challenges of MxIF imaging staining technique: (1) missing stain and (2) missing tissue. Briefly, the missing tissue (upper middle panel) is inevitable during iterative staining and destaining on the same tissue. The missing stain occurs when image quality of a specific channel (stain) is not acceptable, which is an extreme case of missing tissue (occurs 3% of the chance as the empirical observation). The right panel shows the ratio of missing tissue across 9 MxIF stained data. In this work, we focus on the missing stain issue and aim to study 11 structural channels (antibody markers) of MxIF to restore missing stain images.

is a remarkable waste if the subjects with missing stains are excluded from the analysis. Another motivation is to see if there are signal redundancy across the structural stains, which means if we do need so many rounds of stains.

Such scenarios motivate us to develop digital solutions to recover the missing information using deep learning, especially with the image synthesis approaches that have been successfully deployed in various medical imaging applications. The rationale is that the missing signals might be reconstructed by complementary anatomical information, provided by all available signals from the same tissue as shown in Fig. 1.

Generative adversarial networks (GANs) have been broadly validated in medical image synthesis Yu et al. (2020). Prior studies have applied GANs to perform image synthesis in radiology, across the brain Lee et al. (2019); Dar et al. (2019); Zhou et al. (2020); Yu et al. (2019); Zeng and Zheng (2019), thorax Jiang et al. (2018, 2019), abdomen Huo et al. (2018a,b); Yang et al. (2020) and leg Kim et al. (2020). In the past few years, GANs have been applied to histopathology images. For example, Lahiani et al. refined the CycleGAN model to reconstruct H&E whole slide imaging and reduce the tiling artifact Lahiani et al. (2019). Jen et al. utilized patchGAN to convert the routine histochemical stained PAS to multiplex immunohistochemistry stains Jen et al. (2021). Hou et al. used GAN and a task-specific CNN for H&E images synthesis and segmentation masks Hou et al. (2017). Zhang et al. designed a class-conditional neural network to transform two contrast enhanced

unstained tissue autofluorescence images to statin a label-free tissue sample Zhang et al. (2020). Bayramoglu et al. integrated cGAN transforming unstained hyperspectral tissue image to their H&E equivalent Bayramoglu et al. (2017). However, very few, if any, studies have been performed to deal with MxIF stains (modalities). Moreover, the previous methods were trained on relevant small patches (from 256×256 to 384×384 pixel tiles), which would not be ideal for high-resolution pathological images (e.g., 1024×1024).

In this paper, we propose a novel multi-channel high-resolution image synthesis approach, called pixN2N-HD, to tackle any possible combinations of missing stain scenarios in MxIF. Our contribution is three-fold:

1. A single deep network is proposed to tackle missing stain synthesis task in MxIF.

2. The proposed "N-to-N" method saves 2,000-fold computational time (from four years to only 20 hours) compared with training missing stain specific models (e.g., "(N-1)-to-1"), without sacrificing the performance.

3. To our knowledge, this is the first comprehensive experimental study of tackling the missing stain challenge in MxIF via deep synthetic learning.

## 2. Methods

The pix2pix GAN Zhu et al. (2017) is a broadly accepted conditional GAN framework for pixel-wise image style transfer. In the pix2pix design, the discriminator $D$ is trained to distinguish a real and fake image synthesized by the generator $G$. Meanwhile, the generator $G$ is trained to fool the discriminator. The training is performed by using the following GAN loss:

$$L_{GAN}(G, D) = \mathbb{E}_{(x,y)}[\log \mathrm{D}(x,y)] + \mathbb{E}_{(x)}(\log(1 - D(x, G(x))))$$ (1)

where x is the input image, y is the real image. To perform image systhesis on larger images (e.g., >1024×1024 images), a high-resolution version of pix2pix GAN, called pix2pixHD, was invented with two-levels coarse to fine design Wang et al. (2018). However, there is still a technical gap to directly apply pix2pix framework to the missing stain task in MxIF. Briefly, if we directly put all channels as both inputs and outputs, the pix2pix GAN will be degrade to an AutoEncoder, with limited capability to synthesize missing stain. To enable effective N-to-N image synthesis, we introduce the "random gate" (RG) strategy to control the available input and output modalities for the proposed pixN2N-HD GAN. Briefly, 11 input channels and 11 output channels were all used in our network design, where each channel represented one marker.

### 2.1 Missing Stain Synthesis (PixN2N-HD)

The random gate is defined as $\delta$, an 11-dimensional 0-1 binary controller to control the data flow. The overall idea is to formulate available channels ($\delta$ ) as inputs, and simulate missing channels ($\delta$) as outputs. $\delta^{(i)}$ represent the $i^{th}$ channel's gate. When the $\delta^{(i)}$ is turned on, the data in the $i^{th}$ channel will be fed into the generator. Alternatively, if the $\delta^{(i)}$ is turned off, we feed the network with a blank image with all zero values (Fig. 2). Customizing the

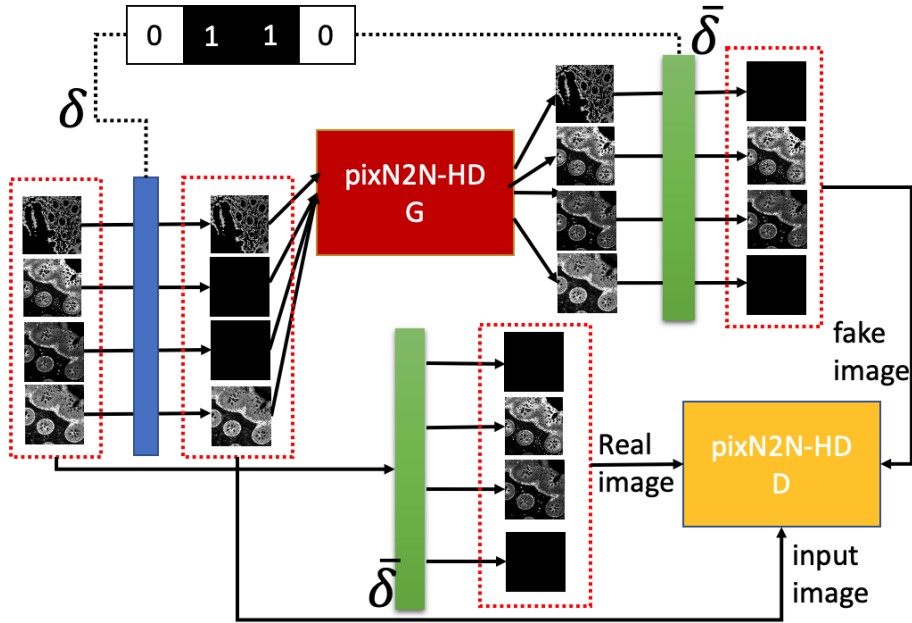

Figure 2: The proposed random gate (RG) strategy in pixN2N-HD is presented. Only 4 markers are shown for illustration. $\delta$ and $\bar{\delta}$ are two inverse gates that controls the data flow in each channel.

"on/off" of $\delta$ provides heterogeneous views of the same data input. For the output side, we reverse the gate $\delta$ to $\bar{\delta}$ to only evaluate the performance of pseudo missing channels. With the random gate, let's set capital letter $M$ indicate that a set of images are used, and $X = \delta(M)$ corresponds to the $x$ in Eq.(1), and $Y = \bar{\delta}(M)$ corresponds to the $y$ in Eq.(1), then the Eq.(1) is generalized to the following formulas:

$$L_{GAN}(G, D) = \mathbb{E}_{(X,Y)}[\log \mathrm{D}(\mathrm{X}, \mathrm{Y})] + \mathbb{E}_X(\log(1 - D(\mathrm{X}, G(\mathrm{X})))) \tag{2}$$

$$\min_G \max_{D1,D2,D3} \sum_{k=1,2,3} L_{GAN}(G, D_k) \tag{3}$$

where $D_k$ is a k level multi-scale discriminator Wang et al. (2018). A feature matching loss (Eq.(4)) is added to stabilize the training of the generator by matching intermediate feature maps in the different layers of the discriminators from real and synthesized images.

$$L_{FM}(G, D_k) = \mathbb{E}_{(\mathrm{X},\mathrm{Y})} \sum_{i=1}^{T} \frac{1}{N_i} \left[ \left\| D_k^{(i)}(\mathrm{X}, \mathrm{Y}) - D_k^{(i)}(\mathrm{X}, G(\mathrm{X})) \right\| \right] \tag{4}$$

where $D_k^{(i)}$ is the $i^{th}$ layer of $D_k$, $N_i$ is the number of elements in each layer and $T$ denotes the total number of layers. Then the final objective function is

$$\min_G \max_{D1,D2,D3} \sum_{k=1,2,3} L_{GAN}(G, D_k) + \lambda L_{FM}(G, D_k) \tag{5}$$

where $\lambda$ controls the weight of GAN loss and feature matching loss.

## 3. Data and Experimental Setting

### 3.0.1 DATASETS.

Nine sample biopsies have been collected from 3 CD patients and 2 healthy controls. Our dataset includes 1 active disease and 3 non-disease biopsies from the ascending colon area, and 5 non-disease biopsies from terminal ileums. The electronic informed consent was obtained from all participants. The protocol was also approved by the Institutional Review Board. The MxIF markers were stained in the following order - DAPI(first round), Muc2, Collagen, $\beta$ catenin, pEGFR, HLA A, PanCK, Na-KATPase, Vimentin, SMA, and $\gamma$Actin. Note that the functional markers were not evaluated in this study since the patterns of the functional markers were more disease dependent. The standard DAPI based registration and auto florescence correction were applied to build pixel to pixel relationship from different markers McKinley et al. (2017). To ensure effective learning, we computed the tissue masks that covered the tissue pixels which contained all markers across all staining rounds. The MxIF data is scanned with 20X magnification, and as a result, the pixels of 9 samples masks were $52,090,325 \pm 21,243,895$ (mean $\pm$ stdev). We applied the masks to each images and preprocessed with group-wise linear normalization.

### 3.0.2 ENVIRONMENTAL SETTING AND IMPLEMENTATION DETAIL.

We randomly chose 4 samples for training, and 5 samples for testing. For the training data, we first split each image into $1024 \times 1024$ patches without re-sampling, and then concatenated the same positions markers' patches together as one data input. During datasets preparation, if any channel contains patch with less than 5% non-zero intensity pixels, we removed the whole patch in the training datasets. Finally, we randomly split 80 percent of the datasets for training (in total 180 patches) and the remaining 20 percent for validation to select the best model epoch. All models were trained by 200 epochs and saved every 10 epochs. The structure similarity index measure (SSIM) was selected to evaluate the synthesis performance. When testing N-to-N and M-to-(N-M) type of models, we synthesized each marker using the validation set and found the epoch with the best average SSIM performance. The SSIM was computed on concatenated patches that have tissues referenced by tissue masks. All experiment models were trained on NVIDIA Titan Xp 12GB graphical card and implemented in PyTorch (`https://github.com/NVIDIA/pix2pixHD`). The coarse G was trained on $512 \times 512$ tile sets with batchSize = 4. The fine G was trained on $1024 \times 1024$ tile sets with batchSize = 1. We chose MSE loss for the GAN loss. In general, each generator's input/output channels only take/generate a specific marker. The input/output channels in our work are set in a sequence of the staining order. Each input/output channel should only take/generate a specific marker. For the random gate setup, as Fig. 2 illustrated, if a marker is randomly chosen as missing in one iteration (gate closed), then the marker's corresponding input channel will be fed with a blank patch, and the corresponding output channel will generate the synthesized marker image. In contrast, if a marker is available (gate open), the marker's relevant output channel output will be wiped blank. Then the loss calculation in the discriminator only considers the synthesized

output of missing stain(s). The other parameters were set by default as described by Wang et al. (2018).

## 4. Results

**Missing stain (missing one stain).** The goal of this experiment is to proof of the concept of the random gate strategy. Fig. 3 presents synthesis results that only one biomarker (stain channel) is missed. All permutations of leave one out synthesis (11 independent "10-to-1" models) and the proposed pixN2N-HD model were evaluated. We evaluated three types of models (1) (N-1)-to-1 synthesis with pix2pixHD to synthesize each marker (stain channel) from all remaining markers, containing 11 independent 10-to-1 models for each missing stain scenario, each model has 10 input channels with 1 output channel; (2) (N-1)-to-1 synthesis with random gate (10-to-1-RG) model was evaluated to repeat the above experiments, again, there are in total of 11 separate models, 10 input channels with 1 output channel, and we random close up to 9 gates on input channels part; and (3) our proposed N-to-N synthesis pixN2N-HD model (a single "11-to-11" model) was evaluated using a single model with 11 input channels and 11 output channels, and we random close up to 10 gates on input channels side. The overall SSIM results (from high to low) of stain-types are in the following order: SMA, Collagen, Na-KATPase, Muc2, pEGFR, Vimentin, HLA A, PanCK, $\gamma$Actin, $\beta$ catenin, DAPI. The Wilcoxon signed-rank test showed that there were no significant differences ($p<0.05$) across any pairs of methods across all markers. The results demonstrated that the single model trained by pixN2N-HD achieved comparable performance relative to the task specific models. This study also showed that the random gate did not harm the performance in (N-1)-to-1 setting.

For the sensitivity test, we trained six pixN2N-HD at the same time with different random gate selection (from random close 1 gate, random close up to 2 gates, up to 3 gates, up to 4 gates, up to 5 gates to no limitation). The average SSIM performance is 0.816 and different model's performance variance is within 2%. And we did not observe significant completion time difference (20 hours), which is similar to train one pix2pixHD model using one NVIDIA Titan Xp 12GB graphic card. Given 11 markers with 2046 possible combinations, training a single pixN2N-HD can potential conserve 2,000-fold computation time (4 years) than state of art (all possible pix2pixHD baseline models) using existing hardware circumstances.

**Missing stain (missing multiple stains).** This experiment aims to validate the scalability of our proposed method to cover different missing stains scenarios. Fig. 4 shows more challenging settings in which more than one stain is missed. We compared the performance between task specific models and a single N-to-N model. Briefly, we directly applied the trained pixN2N-HD model (from **missing one stain**) to synthesize 2 to 5 missing stains. Meanwhile, we train four more separate pix2pixHD models as baselines to cover the same missing stain settings (9-to-2, 8-to-3, 7-to-4 and 6-to-5). The result from Fig. 4 indicates that the random gate did not harm the performance in M-to-(N-M) setting.

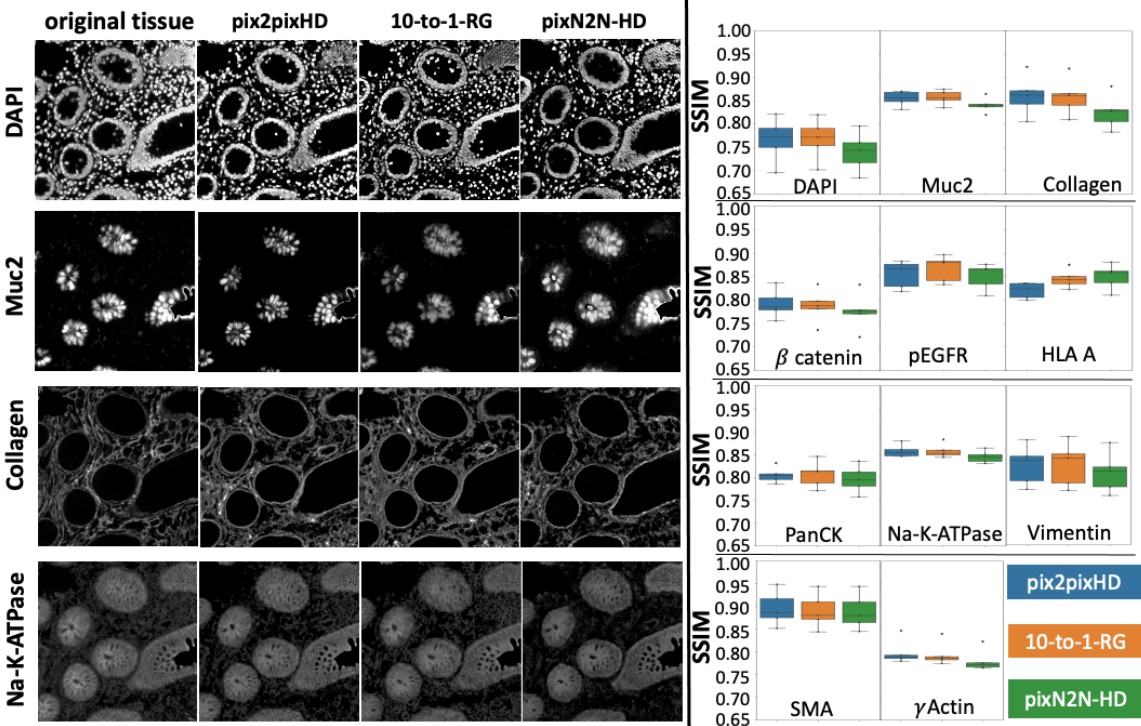

Figure 3: The left panel shows the qualitative synthesis results of having one missing stain channel. The presented patch region of interest is randomly selected with 1024*1024 resolution. The right panel shows the quantitative results with SSIM. Wilcoxon signed-rank test shows the performance across difference methods is not significant.

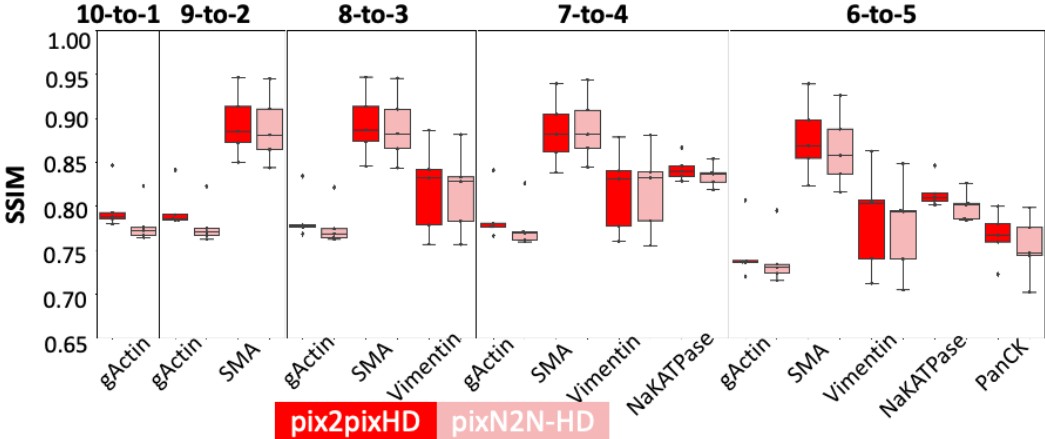

Figure 4: Quantitative results for the missing multiple stains of two models: five M-to-(N-M) use cases between pix2pixHD and pixN2N-HD. The Wilcoxon signed rank test is calculated for each marker's M-to-(N-M) cases without finding significant difference

## 5. Discussion and Conclusion

In this work, we develop a single holistic "N-to-N" conditional GAN to synthesize the missing stain of MxIF imaging. The PixN2N-HD framework with random gate is proposed to enable a single model for all possible missing stain scenarios. The performance of the "N-to-N" model is comparable to the standard "(N-1)-to-1" models (via Pixe2Pix-HD), while requiring 2,000-fold less computational time (from 4 years to 20 hours on one 12 GB GPU) for covering possible permutations of 11 structural stains.

During the MxIF data staining phase, which and what types of markers are failed is unpredictable. Regarding clinical utility, specifically, limited size of tissue samples, such as endoscopic biopsies, and the degradation of the tissues caused by repeated cutting of sections is an impetus for MXIF. Our proposed method has the potential to be generalized and applicable to any other staining protocols (i.e., using different combinations of markers). Moreover, because of the limited sample size, synthetic data generation is a crucial adjunct strategy to complement staining optimization.

The primary purpose of adding the random gate is to refine the model training strategy so that the single N-to-N pix2pixHD model can meet different missing stain scenarios on each training iteration. The data patch size we used is 1024x1024, which is larger than the other histopathology image synthesis studies, as shown in the literature review. There are in total 2,725 patches for training/validation/testing, and we claim that it is sufficient for the training/testing a framework. Our next step is to investigate the batch effect across the MxIF staining multiple batches. For the experiment results, we state that comparing different the SSIM results of different markers is not sufficient. For instance, if a marker contains less tissue with more background (i.e., SMA, Muc2), it might yield higher SSIM, but it does not necessarily mean it is easier to reconstruct than other markers with lower SSIM. So the the fair evaluation should be to compare different methods on each markers individually. Furthermore, the current results cannot demonstrate which stains are redundant.

The most critical future step is to evaluate the performance of real and synthesized images for downstream tasks (e.g., cell segmentation), since that would provide a subjective evaluation of the proposed method compared with SSIM. The next technical step would be to integrate training missing stain and missing tissue synthesis to single multi-task model. Another future direction is to evaluate the performance of the PixN2N-HD against functional markers in MxIF. For instance, we might apply the technique to generate complete membrane masks to assess if it results in more precise segmentation of clumped cells and ultimately creating full cell maps of IBD.

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
