# OpenReview forum: "Random Multi-Channel Image Synthesis for Multiplexed Immunofluorescence Imaging"
_MICCAI.org/2021/Workshop/COMPAY — COMPAY 2021_

### Official Review · Reviewer_4ChE · 2021-08-10
**Potentially interesting work but with limited novelty and experiments**

**Rating:** 6
**Confidence:** 5

**Review:**

The paper proposes an image synthesis approach to predict missing stains in multiplex immunofluorescence images using a generative adversarial network (GAN). Conceptually it draws heavily from the existing pix2pixHD method. The main technical contribution is to use a random gating strategy to control the available input and output channels. Experiments on images from real biopsies illustrate the potential of the method.

Pros:
- Addresses an important problem in cell imaging.
- Introduces random gating for the pix2pixHD method.
- Shows experimental results on real data.

Cons:
- Has limited methodological novelty.
- Is missing a discussion of important related works.
- Uses a very limited dataset in the experiments.

Major comments:

- Related works exists that are not cited but should be discussed and compared with: https://doi.org/10.1038/s41598-020-74500-3, https://doi.org/10.1038/s41592-018-0111-2.

- The presentation of the method in Section 2.1 is rather confusing. It does not seem to match Figure 2. For example, according to Equation (2), D(X,G(X)) is computed, while according to Figure 2 it should be D(Y,$\bar\delta$(G(X))). Also, it is not clear why you would compute D(X,Y), as X and Y are complementary and there is nothing to compare.

- The experiments are based on a very limited dataset, consisting of only nine sample biopsies from three patients and two healthy controls, which makes the conclusions rather premature. This needs to be discussed.

- The first paragraph of Section 4 is supposed to be about "missing one stain" but includes cases which randomly "close up to 9 gates" and "close up to 10 gates", implying that multiple stains may be missing.

Minor comments:

- The citation style must be corrected. For example (page 1 but the same problem happens throughout the manuscript): "bowel inflammation Baumgart and Sandborn (2012)" > "bowel inflammation (Baumgart and Sandborn 2012)".

- Page 1: "occurs 3% of the chance as the empirical observation". Do you mean "empirically observed to occur in 3% of the cases"?

- Page 3: "image systhesis" > "image synthesis".

- Page 4: "gate \delta to \delta" > The latter \delta should have a bar on top.

- Correct the subsection numbering in Section 3.

- Page 5: "auto florescence" > "autofluorescence".

- Page 6: "to proof of the concept" > "to provide a proof of concept".

---

### Official Review · Reviewer_MGJe · 2021-08-18
**Review by MGJe**

**Rating:** 6
**Confidence:** 3

**Review:**

The paper proposes to enhance MxIF images by addressing artefacts related to missing tissue/stain regions caused by repeated staining and bleaching using generative modelling. They derive from pixel level region generation ideas such as pix2pix to build a conditional GAN based approach for filling in depleted tissue regions across multiple channels in multiplexed immunofluroroscence imaging. This method does have the potential to address an important operational problem in IF imaging, as depleted tissue in imaging can cause significant loss of detail during downstream analysis.

That said, the paper has the following weaknesses in my opinion:
1. Concepts such as the N-to-N approach are inadequately described and more generally, there are a number of concepts introduced with a multiplicity of terminology that makes it a somewhat confusing article to read. A suggestion would be to use graphical flow charts to describe the sequences used for each of the experiments.
2. How are the issues of consistency across channels addressed? Also, what is the sensitivity of the generative modules to the extent of depletion of tissues or missing stain?
3. What are the key pitfalls of the proposed approach? The future work sections mention extension to downstream tasks, and integration of the missing stain and missing tissue synthesis stages to single multi-task model. It would be good to understand the failure modes of the approach so that one can appreciate how robust the model may be upon deployment to downstream tasks.

---

### Decision · Program_Chairs · 2021-08-25

Accept